# Sample Complexity of Learning Heuristic Functions for Greedy-Best-First and A* Search

**Shinsaku Sakaue**
The University of Tokyo
Tokyo, Japan
sakaue@mist.i.u-tokyo.ac.jp

**Taihei Oki**
The University of Tokyo
Tokyo, Japan
oki@mist.i.u-tokyo.ac.jp

## Abstract

Greedy best-first search (GBFS) and A* search (A*) are popular algorithms for path-finding on large graphs. Both use so-called heuristic functions, which estimate how close a vertex is to the goal. While heuristic functions have been handcrafted using domain knowledge, recent studies demonstrate that learning heuristic functions from data is effective in many applications. Motivated by this emerging approach, we study the sample complexity of learning heuristic functions for GBFS and A*. We build on a recent framework called *data-driven algorithm design* and evaluate the *pseudo-dimension* of a class of utility functions that measure the performance of parameterized algorithms. Assuming that a vertex set of size $n$ is fixed, we present $O(n \lg n)$ and $O(n^2 \lg n)$ upper bounds on the pseudo-dimensions for GBFS and A*, respectively, parameterized by heuristic function values. The upper bound for A* can be improved to $O(n^2 \lg d)$ if every vertex has a degree of at most $d$ and to $O(n \lg n)$ if edge weights are integers bounded by $\mathrm{poly}(n)$. We also give $\Omega(n)$ lower bounds for GBFS and A*, which imply that our bounds for GBFS and A* under the integer-weight condition are tight up to a $\lg n$ factor. Finally, we discuss a case where the performance of A* is measured by the suboptimality and show that we can sometimes obtain a better guarantee by combining a parameter-dependent worst-case bound with a sample complexity bound.

## 1 Introduction

Given a graph with a start vertex $s$, a goal vertex $t$, and non-negative edge weights, we consider finding an $s$–$t$ path with a small total weight. The Dijkstra algorithm [16] finds an optimal path by exploring all vertices that are as close to $s$ as $t$. It, however, is sometimes impractical for large graphs since exploring all such vertices is too costly. Heuristic search algorithms are used to address such situations; among them, greedy best-first search (GBFS) [17] and A* search (A*) [24] are two popular algorithms. Both GBFS and A* use so-called heuristic functions, which estimate how close an input vertex is to $t$. GBFS/A* attempts to avoid redundant exploration by scoring vertices based on heuristic function values and iteratively expanding vertices with the smallest score. If well-suited heuristic functions are available, GBFS/A* can run much faster than the Dijkstra algorithm. Furthermore, if A* uses an *admissible* heuristic function, i.e., it never overestimates the shortest-path distance to $t$, it always finds an optimal path [24]. Traditionally, heuristic functions have been made based on domain knowledge; e.g., if graphs are road networks, the Euclidean distance gives an admissible heuristic.

When applying GBFS/A* to various real-world problems, a laborious process is to handcraft heuristic functions. Learning heuristic functions from data can be a promising approach to overcoming the obstacle due to the recent development of technologies for collecting graph data. Researchers have demonstrated the effectiveness of this approach in robotics [11, 32, 28, 36], computational organic chemistry [13], and pedestrian trajectory prediction [36]. With learned heuristic functions, however,

obtaining theoretical guarantees is difficult since we can hardly understand how the search can be guided by such heuristic functions. (A recent paper [1] studies learning of admissible heuristics for A*, but the optimality is confirmed only empirically.) Moreover, learned heuristic functions may be overfitting to problem instances at hand. That is, even if GBFS/A* with learned heuristic functions perform well over training instances, they may deliver poor future performance. In summary, the emerging line of work on search algorithms with learned heuristic functions is awaiting a theoretical foundation for guaranteeing their performance in a data-driven manner. Thus, a natural question is: *how many sampled instances are needed to learn heuristic functions with generalization guarantees on the performance of resulting GBFS/A*?*

## 1.1 Our contribution

We address the above question, assuming that path-finding instances defined on a fixed vertex set of size $n$ are drawn i.i.d. from an unknown distribution. Our analysis is based on so-called *data-driven algorithm design* [22, 4], a PAC-learning framework for bounding the sample complexity of algorithm configuration. In the analysis, the most crucial step is to evaluate the *pseudo-dimension* of a class of utility functions that measure the performance of parameterized algorithms. We study the case where GBFS/A* is parameterized by heuristic function values and make the following contributions:

1. Section 3 gives $O(n \lg n)$ and $O(n^2 \lg n)$ upper bounds on the pseudo-dimensions for GBFS and A*, respectively. The bound for A* can be improved to $O(n^2 \lg d)$ if every vertex has an at most $d$ degree and to $O(n \lg n)$ if edge weights are non-negative integers at most $\mathrm{poly}(n)$.

2. Section 4 presents $\Omega(n)$ lower bounds on the pseudo-dimensions for GBFS and A*. We prove this result by constructing $\Omega(n)$ instances with unweighted graphs. Thus, our bounds for GBFS and A* under the integer edge-weight condition are tight up to a $\lg n$ factor.

3. Section 5 studies a particular case of bounding the suboptimality of A*. We show that we can sometimes improve the guarantee obtained in Section 3 by using an alternative $O(n \lg n)$ bound on the pseudo-dimension of a class of parameter-dependent worst-case bounds [34].

An important consequence of the above results is the tightness up to a $\lg n$ factor for GBFS and A* under the integer-weight assumption. Note that this assumption holds in various realistic situations. For example, the Internet network and state-space graphs of games are unweighted (unit-weight) graphs, and A* is often applied to path-finding instances on such graphs.

## 1.2 Related work

**Data-driven algorithm design.** Gupta and Roughgarden [22] proposed a PAC approach for bounding the sample complexity of algorithm configuration, which is called *data-driven algorithm design* and has been applied to a broad family of algorithms, including greedy, clustering, and sequence alignment algorithms. We refer the reader to a nice survey [4]. A recent line of work [5, 9, 10] has extensively studied the sample complexity of configuring integer-programming methods, e.g., branch-and-bound and branch-and-cut. In [9, 10], upper bounds on the pseudo-dimension for general tree search are presented, which are most closely related to our results. Our upper bounds, which are obtained by using specific properties of GBFS/A*, are better than the previous bounds for general tree search, as detailed in Appendix A. Balcan et al. [8] presented a general framework for evaluating the pseudo-dimension. Their idea is to suppose that performance measures form a class of functions of algorithm parameters, called *dual* functions, and characterize its complexity based on how they are piecewise structured. This idea plays a key role in the analysis of [9, 10], and our analysis of the upper bounds are also inspired by their idea. Its application to our setting, however, requires a close look at the behavior of GBFS/A*. Balcan et al. [7] showed that approximating dual functions with simpler ones is useful for improving sample complexity bounds, which is similar to our idea in Section 5. A difference is that while they construct simpler functions with a dynamic programming algorithm, we can use a known worst-case bound on the suboptimality of best-first search [34]. Lower bounds on the pseudo-dimension for graph-search algorithms have not been well studied.

**Heuristic search with learning.** Eden et al. [18] theoretically studied how the average-case running time of A* can be affected by the dimensions or bits of learned embeddings or labels of vertex features, based on which heuristic function values and computed. The sample complexity of learning heuristic functions, however, has not been studied.

## 2 Preliminaries

We present the background on learning theory and our problem setting. In what follows, we let $\mathbb{I}(\cdot)$ be a boolean function that returns 1 if its argument is true and 0 otherwise. We use $\mathcal{H} \subseteq \mathcal{R}^{\mathcal{Y}}$ to denote a class of functions that map $\mathcal{Y}$ to $\mathcal{R} \subseteq \mathbb{R}$. For any positive integer $m$, we let $[m] = \{1, \ldots, m\}$.

### 2.1 Background on learning theory

The following *pseudo-dimension* [29] is a fundamental notion for quantifying the complexity of a class of real-valued functions.

**Definition 1.** Let $\mathcal{H} \subseteq \mathbb{R}^{\mathcal{Y}}$ be a class of functions that map some domain $\mathcal{Y}$ to $\mathbb{R}$. We say a set $\{y_1, \ldots, y_N\} \subseteq \mathcal{Y}$ is *shattered* by $\mathcal{H}$ if there exist target values, $t_1, \ldots, t_N \in \mathbb{R}$, such that

$$|\{(\mathbb{I}(h(y_1) \geq t_1), \ldots, \mathbb{I}(h(y_N) \geq t_N)) \mid h \in \mathcal{H}\}| = 2^N.$$

The *pseudo-dimension* of $\mathcal{H}$, denoted by $\mathrm{Pdim}(\mathcal{H})$, is the size of a largest set shattered by $\mathcal{H}$.

If $\mathcal{H}$ is a set of binary-valued functions that map $\mathcal{Y}$ to $\{0, 1\}$, the pseudo-dimension of $\mathcal{H}$ coincides with the so-called *VC-dimension* [35], which is denoted by $\mathrm{VCdim}(\mathcal{H})$.

The following proposition enables us to obtain sample complexity bounds by evaluating the pseudo-dimension (see, e.g., [2, Theorem 19.2] and [27, Theorem 11.8]).

**Proposition 1.** *Let $H > 0$, $\mathcal{H} \subseteq [0, H]^{\mathcal{Y}}$, and $\mathcal{D}$ be a distribution over $\mathcal{Y}$. For any $\delta \in (0, 1)$, with a probability of at least $1 - \delta$ over the i.i.d. draw of $\{y_1, \ldots, y_N\} \sim \mathcal{D}^N$, for all $h \in \mathcal{H}$, it holds that*

$$\left| \frac{1}{N} \sum_{i=1}^{N} h(y_i) - \mathbb{E}_{y \sim \mathcal{D}}[h(y)] \right| = O\left( H \sqrt{\frac{\mathrm{Pdim}(\mathcal{H}) \lg \frac{N}{\mathrm{Pdim}(\mathcal{H})} + \lg \frac{1}{\delta}}{N}} \right).$$

In other words, for any $\epsilon > 0$, $N = \Omega\left( \frac{H^2}{\epsilon^2} \left( \mathrm{Pdim}(\mathcal{H}) \lg \frac{H}{\epsilon} + \lg \frac{1}{\delta} \right) \right)$ sampled instances are sufficient to ensure that with a probability of at least $1 - \delta$, for all $h \in \mathcal{H}$, the difference between the empirical average and the expectation over an unknown distribution $\mathcal{D}$ is at most $\epsilon$.

### 2.2 Problem formulation

We describe path-finding instances, GBFS/A* algorithm, and performance measures considered in this paper.

**Path-finding instances.**    We consider solving randomly generated path-finding instances repetitively. Let $x = (V, E, \{w_e\}_{e \in E}, s, t)$ be a path-finding instance, where $(V, E)$ is a simple directed graph with $n$ vertices, $\{w_e\}_{e \in E}$ is a set of non-negative edge weights (sometimes called costs), $s \in V$ is a start vertex, and $t \in V$ is a goal vertex. We let $\Pi$ be a class of possible instances. Each instance $x \in \Pi$ is drawn from an unknown distribution $\mathcal{D}$ over $\Pi$. We impose the following assumption on $\Pi$.

**Assumption 1.** *For all $x \in \Pi$, the vertex set $V$ and the goal node $t$ are identical, and there always exists at least one directed path from $s \neq t$ to $t$, i.e., every instance $x \in \Pi$ is feasible.*

Fixing $V$ is necessary for evaluating the pseudo-dimension in terms of $n = |V|$. Note that we can deal with the case where some instances in $\Pi$ are defined on vertex subsets $V' \subseteq V$ by removing edges adjacent to $V \setminus V'$. The feasibility assumption is needed to ensure that GBFS/A* always returns a solution, and $s \neq t$ simply rules out the trivial case where the empty set is optimal. In Appendix B, we discuss how to extend our results to the case where $t$ can change depending on instances.

**Algorithm description.**    We sketch algorithmic procedures that are common to both GBFS and A* (see Algorithms 1 and 2 for details, respectively). Let $A_{\boldsymbol{\rho}}$ be a GBFS/A* algorithm, which is parameterized by heuristic function values $\boldsymbol{\rho} \in \mathbb{R}^n$. Given an instance $x \in \Pi$, $A_{\boldsymbol{\rho}}$ starts from $s$ and iteratively builds a set of candidate paths. These paths are maintained by OPEN and CLOSED lists, together with pointers $\mathrm{p}(\cdot)$ to parent vertices. The OPEN list contains vertices to be explored, and the CLOSED list consists of vertices that have been explored. In each iteration, we select a vertex $v$ from OPEN, expand $v$, and move $v$ from OPEN to CLOSED.

Heuristic function values $\rho$ are used when selecting vertices. For each $v \in V$, the corresponding entry in $\rho$, denoted by $\rho_v$, represents an estimated shortest-path distance from $v$ to $t$. (Although heuristic function values are usually denoted by $h(v)$, we here use $\rho_v$ for convenience.) In each iteration, we select a vertex with the smallest *score*, which is defined based on $\rho$ as detailed later. We impose the following assumption on the vertex selection step.

**Assumption 2.** *Define an arbitrary strict total order on $V$; for example, we label elements in $V$ by $v_1, \ldots, v_n$ and define a total order $v_1 < \cdots < v_n$. When selecting a vertex with the smallest score, we break ties, if any, in favor of the smallest vertex with respect to the total order.*

If we allow $A_\rho$ to break ties arbitrarily, its behavior becomes too complex to obtain meaningful bounds on the pseudo-dimension. Assumption 2 is a natural rule to exclude such troublesome cases.

**Performance measure.** Let $A_\rho$ be GBFS/A* with parameters $\rho \in \mathbb{R}^n$. We measure performance of $A_\rho$ on $x \in \Pi$ with a utility function $u$. We assume $u$ to satisfy the following condition.

**Assumption 3.** *Let $H > 0$. A utility function $u$ takes $x$ and a series of all* OPEN, CLOSED, *and* $\mathrm{p}(\cdot)$ *generated during the execution of $A_\rho$ on $x \in \Pi$ as input, and returns a scalar value in $[0, H]$.*

We sometimes use $A_\rho$ to represent the series of OPEN and CLOSED lists and pointers generated by $A_\rho$. Note that $u$ meeting Assumption 3 can measure various kinds of performance. For example, since the pointers indicate an $s$–$t$ path returned by $A_\rho$, $u$ can represent its cost. Moreover, since the series of OPEN and CLOSED lists maintain all search states, $u$ can represent the time and space complexity of $A_\rho$. We let $u_\rho : \Pi \to [0, H]$ denote the utility function that returns the performance of $A_\rho$ on any $x \in \Pi$, and define a class of such functions as $\mathcal{U} = \{ u_\rho : \Pi \to [0, H] \mid \rho \in \mathbb{R}^n \}$. The upper bound, $H$, is necessary to obtain sample complexity bounds with Proposition 1. Setting such an upper bound is usual in practice. For example, if $u$ measures the running time, $H$ represents a time-out deadline.

**Generalization guarantees on performance.** Given the above setting, we want to learn $\hat{\rho}$ values that attain an optimal $\mathbb{E}_{x \sim \mathcal{D}}[u_{\hat{\rho}}(x)]$ value, where available information consists of sampled instances $x_1, \ldots, x_N$ and $u_\rho(x_1), \ldots, u_\rho(x_N)$ values for any $\rho \in \mathbb{R}^n$. To obtain generalization guarantees on the performance of $A_{\hat{\rho}}$, we bound $|\frac{1}{N} \sum_{i=1}^{N} u_\rho(x_i) - \mathbb{E}_{x \sim \mathcal{D}}[u_\rho(x)]|$ uniformly for all $\rho \in \mathbb{R}^n$. Note that the uniform bound offers performance guarantees that are independent of learning procedures, e.g., manual or automated (without being uniform, learned $\hat{\rho}$ may be overfitting sampled instances). As in Proposition 1, to bound the sample complexity of learning $\rho$ values, we need to evaluate the pseudo-dimension of $\mathcal{U}$, denoted by $\mathrm{Pdim}(\mathcal{U})$, which is the main subject of this study.

**Remarks on heuristic functions.** While we allow heuristic function values $\rho$ to be any point in $\mathbb{R}^n$, the range of heuristic functions may be restricted to some subspace of $\mathbb{R}^n$. Note that our upper bounds are applicable to such situations since restricting the space of possible $\rho$ values does not increase $\mathrm{Pdim}(\mathcal{U})$. Meanwhile, such restriction may be useful for improving the upper bounds on $\mathrm{Pdim}(\mathcal{U})$; exploring this direction is left for future work. Also, our setting cannot deal with heuristic functions that take some instance-dependent features as input. To study such cases, we need more analysis that is specific to heuristic function models, which goes beyond the scope of this paper. Thus, we leave this for future work. Note that our setting still includes important heuristic function models on fixed vertex sets. For example, we can set $\rho$ using learned distances to landmarks [21], or we can let $\rho$ be distances measured on some metric space by learning metric embeddings of vertices [37].

## 3 Upper bounds on the pseudo-dimension

We present details of GBFS and A* and upper bounds on the pseudo-dimensions of $\mathcal{U}$. In this section, we suppose that vertices in $V$ are labeled by $v_1, \ldots, v_n$ as in Assumption 2.

### 3.1 Greedy best-first search

Algorithm 1 shows the details of GBFS $A_\rho$ with heuristic function values $\rho \in \mathbb{R}^n$. When selecting vertices in Step 3, the scores are determined only by $\rho$. This implies an obvious but important fact.

**Lemma 1.** *Let $\rho, \rho' \in \mathbb{R}^n$ be a pair of heuristic function values with an identical total order up to ties on their entries, i.e., $\mathbb{I}(\rho_{v_i} \le \rho_{v_j}) = \mathbb{I}(\rho'_{v_i} \le \rho'_{v_j})$ for all $i, j \in [n]$ such that $i < j$. Then, we have $u_\rho(x) = u_{\rho'}(x)$ for all $x \in \Pi$.*

---

**Algorithm 1** GBFS with heuristic function values $\boldsymbol{\rho}$

---

1: OPEN $= \{s\}$, CLOSED $= \emptyset$, and $\mathrm{p}(s) = $ None.
2: **while** OPEN is not empty **:**
3:     $v \leftarrow \operatorname{argmin}\{ \rho_{v'} \mid v' \in$ OPEN $\}$.                ▷ Break ties as in Assumption 2.
4:     **for** each child $c$ of $v$ **:**
5:        **if** $c = t$ **:**
6:           **return** $s$–$t$ path by tracing pointers $\mathrm{p}(\cdot)$, where $\mathrm{p}(t) = v$.
7:        **if** $c \notin$ OPEN $\cup$ CLOSED **:**
8:           $\mathrm{p}(c) \leftarrow v$ and OPEN $\leftarrow$ OPEN $\cup \{c\}$.
9:     Move $v$ from OPEN to CLOSED.

---

---

**Algorithm 2** A* with heuristic function values $\boldsymbol{\rho}$

---

1: OPEN $= \{s\}$, CLOSED $= \emptyset$, $\mathrm{p}(s) = $ None, and $g_s = 0$.
2: **while** OPEN is not empty **:**
3:     $v \leftarrow \operatorname{argmin}\{ g_{v'} + \rho_{v'} \mid v' \in$ OPEN $\}$.          ▷ Break ties as in Assumption 2.
4:     **if** $v = t$ **:**
5:        **return** $s$–$t$ path by tracing pointers $\mathrm{p}(\cdot)$.
6:     **for** each child $c$ of $v$ **:**
7:        $g_{\mathrm{new}} \leftarrow g_v + w_{(v,c)}$.
8:        **if** $c \notin$ OPEN $\cup$ CLOSED **:**
9:           $g_c \leftarrow g_{\mathrm{new}}$, $\mathrm{p}(c) \leftarrow v$, and OPEN $\leftarrow$ OPEN $\cup \{c\}$.
10:       **else if** $c \in$ OPEN and $g_{\mathrm{new}} < g_c$ **:**
11:          $g_c \leftarrow g_{\mathrm{new}}$ and $\mathrm{p}(c) \leftarrow v$.
12:       **else if** $c \in$ CLOSED and $g_{\mathrm{new}} < g_c$ **:**       ▷ Steps 12–14 are for reopening.
13:          $g_c \leftarrow g_{\mathrm{new}}$ and $\mathrm{p}(c) \leftarrow v$.
14:          Move $c$ from CLOSED to OPEN.
15:     Move $v$ from OPEN to CLOSED.

---

*Proof.* For any $x \in \Pi$, if $\boldsymbol{\rho}$ and $\boldsymbol{\rho}'$ have an identical strict total order on their entries, vertices selected in Step 3 are the same in each iteration of $A_{\boldsymbol{\rho}}$ and $A_{\boldsymbol{\rho}'}$. Since this is the only step $\boldsymbol{\rho}$ and $\boldsymbol{\rho}'$ can affect, we have $A_{\boldsymbol{\rho}} = A_{\boldsymbol{\rho}'}$ for all $x \in \Pi$, hence $u_{\boldsymbol{\rho}}(x) = u_{\boldsymbol{\rho}'}(x)$. Moreover, this holds even if $\boldsymbol{\rho}$ and/or $\boldsymbol{\rho}'$ have ties on their entries because of Assumption 2. That is, the total order uniquely determines a vertex selected in Step 3 even in case of ties. Therefore, the statement holds. □

From Lemma 1, the behavior of GBFS is uniquely determined once a total order on $\{\rho_v\}_{v \in V}$ is fixed. Thus, for any $x \in \Pi$, the number of distinct $u_{\boldsymbol{\rho}}(x)$ values is at most $n!$, the number of permutations of $\{\rho_v\}_{v \in V}$. This fact enables us to obtain an $\mathrm{O}(n \lg n)$ upper bound on the pseudo-dimension of $\mathcal{U}$.

**Theorem 1.** *For GBFS $A_{\boldsymbol{\rho}}$ with parameters $\boldsymbol{\rho} \in \mathbb{R}^n$, it holds that* $\mathrm{Pdim}(\mathcal{U}) = \mathrm{O}(n \lg n)$.

*Proof.* Lemma 1 implies that we can partition $\mathbb{R}^n$ into $n!$ regions, $\mathcal{P}_1, \mathcal{P}_2, \ldots$, so that for every $\mathcal{P}_i$, any pair of $\boldsymbol{\rho}, \boldsymbol{\rho}' \in \mathcal{P}_i$ satisfies $u_{\boldsymbol{\rho}}(x) = u_{\boldsymbol{\rho}'}(x)$ for all $x \in \Pi$. Note that the construction of the regions, $\mathcal{P}_1, \mathcal{P}_2, \ldots$, does not depend on $x$. Thus, given any $N$ instances $x_1, \ldots, x_N$, even if $\boldsymbol{\rho}$ moves over whole $\mathbb{R}^n$, the number of distinct tuples of form $(u_{\boldsymbol{\rho}}(x_1), \ldots, u_{\boldsymbol{\rho}}(x_N))$ is at most $n!$. To shatter $N$ instances, $n! \geq 2^N$ must hold. Solving this for the largest $N$ yields $\mathrm{Pdim}(\mathcal{U}) = \mathrm{O}(n \lg n)$. □

### 3.2 A* search

Algorithm 2 is the details of A*. As with GBFS, $\boldsymbol{\rho}$ only affects the vertex selection step (Step 3). However, unlike GBFS, the scores, $g_v + \rho_v$, involve not only $\boldsymbol{\rho}$ but also $\{g_v\}_{v \in V}$. Each $g_v$ is called a $g$-cost and maintains a cost of some path from $s$ to $v$. As in Algorithm 2, when $v$ is expanded and a shorter path to $c$ is found, whose cost is denoted by $g_{\mathrm{new}}$, we update the $g_c$ value. Thus, each $g_v$ always gives an upper bound on the shortest-path distance from $s$ to $v$. For each $v \in V$, there are at most $\sum_{k=0}^{n-2} k! \leq (n-1)!$ simple paths connecting $s$ to $v$, and thus $g_v$ can take at most $(n-1)!$ distinct values. We denote the set of those distinct values by $\mathcal{G}_v$, and define $\mathcal{G}_V = \{ (v, g_v) \mid v \in V, g_v \in \mathcal{G}_v \}$ as the set of all pairs of a vertex and its possible $g$-cost. It holds that $|\mathcal{G}_V| \leq n \times (n-1)! = n!$.

Note that once $x \in \Pi$ is fixed, $\mathcal{G}_v$ for $v \in V$ and $\mathcal{G}_V$ are uniquely determined. To emphasize this fact, we sometimes use notation with references to $x$: $g_v(x)$, $\mathcal{G}_v(x)$, and $\mathcal{G}_V(x)$. As with the case of GBFS (Lemma 1), we can define a total order on the scores to determine the behavior of A* uniquely.

**Lemma 2.** *Fix any instance $x \in \Pi$. Let $\boldsymbol{\rho}, \boldsymbol{\rho}' \in \mathbb{R}^n$ be a pair of heuristic function values such that total orders on the sets of all possible scores, $\{ g_v(x) + \rho_v \mid (v, g_v(x)) \in \mathcal{G}_V(x) \}$ and $\{ g_v(x) + \rho'_v \mid (v, g_v(x)) \in \mathcal{G}_V(x) \}$, are identical up to ties. Then, it holds that $u_{\boldsymbol{\rho}}(x) = u_{\boldsymbol{\rho}'}(x)$.*

*Proof.* If the two sets of scores have an identical strict total order, we select the same vertex in Step 3 in each iteration of $A_{\boldsymbol{\rho}}$ and $A_{\boldsymbol{\rho}'}$. Thus, we have $A_{\boldsymbol{\rho}} = A_{\boldsymbol{\rho}'}$ for any fixed $x$, implying $u_{\boldsymbol{\rho}}(x) = u_{\boldsymbol{\rho}'}(x)$. We show that this holds even in the presence of ties by using Assumption 2. First, any two scores of the same vertices, $g_v(x) + \rho_v$ and $g'_v(x) + \rho_v$, never have ties since $\mathcal{G}_v$ consists of distinct $g$-costs. Next, if $g_{v_i}(x) + \rho_{v_i} = g_{v_j}(x) + \rho_{v_j}$ holds for some $i < j$, we always prefer $v_i$ to $v_j$ in Step 3 due to Assumption 2. Therefore, even in the presence of ties, we select a vertex in Step 3 as if the set of scores has a strict total order. Thus, if $\boldsymbol{\rho}$ and $\boldsymbol{\rho}'$ induce the same total order up to ties on the sets of possible scores, it holds that $u_{\boldsymbol{\rho}}(x) = u_{\boldsymbol{\rho}'}(x)$. $\square$

By using Lemma 2, we can obtain an $O(n^2 \lg n)$ upper bound on the pseudo-dimension of $\mathcal{U}$.

**Theorem 2.** *For A* $A_{\boldsymbol{\rho}}$ with parameters $\boldsymbol{\rho} \in \mathbb{R}^n$, it holds that $\mathrm{Pdim}(\mathcal{U}) = O(n^2 \lg n)$.*

*Proof.* As with the proof of Theorem 1, we partition $\mathbb{R}^n$ into some regions so that in each region, the behavior of A* is unique. Unlike the case of GBFS, boundaries of such regions change over $N$ instances. To deal with this situation, we use a geometric fact: for $m \geq n \geq 1$, $m$ hyperplanes partition $\mathbb{R}^n$ into $O((em)^n)$ regions.[1]

Fix a tuple of any $N$ instances $(x_1, \dots, x_N)$. We consider hyperplanes in $\mathbb{R}^n$ of form $g_{v_i}(x_k) + \rho_{v_i} = g_{v_j}(x_k) + \rho_{v_j}$ for all $k \in [N]$ and all pairs of $(v_i, g_{v_i}(x_k)), (v_j, g_{v_j}(x_k)) \in \mathcal{G}_V$ such that $i \neq j$. These hyperplanes partition $\mathbb{R}^n$ into some regions, $\mathcal{P}_1, \mathcal{P}_2, \dots$, so that the following condition holds: for every $\mathcal{P}_i$, any $\boldsymbol{\rho}, \boldsymbol{\rho}' \in \mathcal{P}_i$ have the same total order on $\{ g_v(x_k) + \rho_v \mid (v, g_v(x_k)) \in \mathcal{G}_V(x) \}$ and $\{ g_v(x_k) + \rho'_v \mid (v, g_v(x_k)) \in \mathcal{G}_V(x_k) \}$ up to ties for all $k \in [N]$, which implies $u_{\boldsymbol{\rho}}(x_k) = u_{\boldsymbol{\rho}'}(x_k)$ for all $k \in [N]$ due to Lemma 2. That is, for every $k \in [N]$, if we see $u_{\boldsymbol{\rho}}(x_k)$ as a function of $\boldsymbol{\rho}$, it is piecewise constant where pieces are given by $\mathcal{P}_1, \mathcal{P}_2, \dots$. Therefore, when $\boldsymbol{\rho}$ moves over whole $\mathbb{R}^n$, the number of distinct tuples of form $(u_{\boldsymbol{\rho}}(x_1), \dots, u_{\boldsymbol{\rho}}(x_N))$ is at most the number of the pieces. Note that the pieces are generated by partitioning $\mathbb{R}^n$ with $\sum_{k \in [N]} \binom{|\mathcal{G}_V(x_k)|}{2} \leq N \binom{n!}{2}$ hyperplanes, which means there are at most $O\left( \left( eN \binom{n!}{2} \right)^n \right)$ pieces. To shatter $N$ instances, $O\left( \left( eN \binom{n!}{2} \right)^n \right) \geq 2^N$ is necessary. Solving this for the largest $N$ yields $\mathrm{Pdim}(\mathcal{U}) = O(n^2 \lg n)$. $\square$

Compared with GBFS, the additional $n$ factor comes from the bound of $(n-1)!$ on $|\mathcal{G}_v|$. This bound may seem too pessimistic, but it is almost tight in some cases, as implied by the following example.

**Example 1.** Let $(V, E)$ be a complete graph with edges labeled as $\{e_1, \dots, e_{|E|}\}$. Set each edge weight $w_{e_i}$ to $2^{i-1}$ for $i \in [|E|]$. Considering the binary representation of the edge weights, the costs of all simple $s$–$v$ paths are mutually different for $v \in V$, which implies $|\mathcal{G}_v| = \sum_{k=0}^{n-2} k! \geq (n-2)!$.

This example suggests that improving the $O(n^2 \lg n)$ bound is not straightforward. Under some realistic assumptions, however, we can improve it by deriving smaller upper bounds on $|\mathcal{G}_v|$.

First, if the maximum degree of vertices is always bounded, we can obtain the following bound.

**Theorem 3.** *Assume that the maximum out-degrees of directed graphs $(V, E)$ of all instances in $\Pi$ are upper bounded by $d$. Then, it holds that $\mathrm{Pdim}(\mathcal{U}) = O(n^2 \lg d)$.*

*Proof.* Under the assumption on the maximum degree, there are at most $\sum_{k=0}^{n-2} d^k \leq (n-1)d^{n-2}$ simple $s$–$v$ paths, which implies $|\mathcal{G}_v| \leq (n-1)d^{n-2}$ for every $v \in V$. Therefore, we have $|\mathcal{G}_V| \leq n \times (n-1)d^{n-2}$. Following the proof of Theorem 2, we can obtain an upper bound on $\mathrm{Pdim}(\mathcal{U})$ by solving $O\left( N^n \binom{n(n-1)d^{n-2}}{2}^n \right) \geq 2^N$ for the largest $N$, which yields $\mathrm{Pdim}(\mathcal{U}) = O(n^2 \lg d)$. $\square$

---

[1]Even if some regions degenerate, from [23, Theorem 28.1.1] and [12, Proposition A2.1], the number of all $d$-dimensional regions for $d = 0, \dots, n$ is $\sum_{d=0}^{n} \sum_{i=0}^{d} \binom{n-i}{d-i} \binom{m}{n-i} \leq 2(em)^n$. The fact has a close connection to Sauer's lemma [30] (see [20]). In this sense, our analysis is in a similar spirit to the general framework of [8].

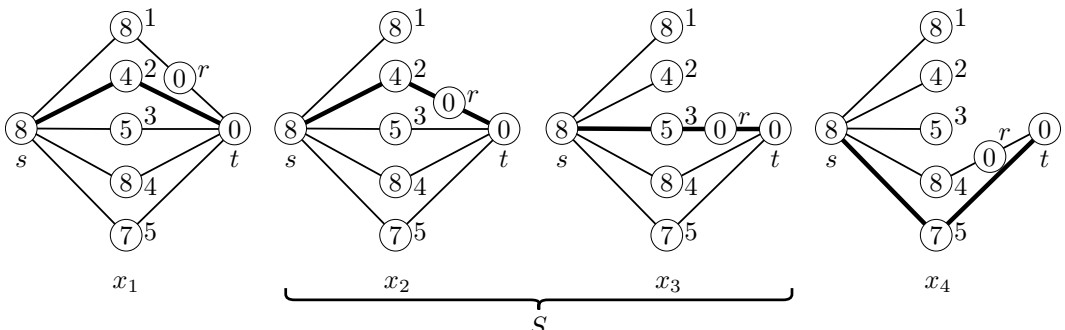

Figure 1: An illustration of the instances $x_1, \ldots, x_{n-4}$ for $n = 8$. Each vertex is labeled by $s$, $r$, $t$, or $i \in [n-3]$, as shown nearby the vertex circles. The values in vertex circles represent $\rho$ that makes $A_\rho$ return suboptimal paths to $x_2$ and $x_3$, i.e., $S = \{2, 3\}$. The thick edges indicate returned paths.

Second, if edge weights are non-negative integers bounded by $W$, we can obtain the following bound.

**Theorem 4.** *Assume that edge weights $\{w_e\}_{e \in E}$ of all instances in $\Pi$ are non-negative integers bounded by a constant $W$ from above. Then, it holds that $\mathrm{Pdim}(\mathcal{U}) = \mathrm{O}(n \lg(nW))$.*

*Proof.* Under the assumption, every $g$-cost $g_v$ takes a non-negative integer value at most $nW$. Since $\mathcal{G}_v$ consists of distinct $g$-cost values, $|\mathcal{G}_v| \le nW$ holds, hence $|\mathcal{G}_V| \le n^2 W$. Solving $\mathrm{O}\left(N^n \binom{n^2 W}{2}^n\right) \ge 2^N$ for the largest $N$, we obtain $\mathrm{Pdim}(\mathcal{U}) = \mathrm{O}(n \lg(nW))$. $\qquad \square$

Note that if $W = \mathrm{O}(\mathrm{poly}(n))$ holds, we have $\mathrm{Pdim}(\mathcal{U}) = \mathrm{O}(n \lg n)$.

**On reopening.** A* is usually allowed to reopen closed vertices as in Steps 12–14. This, however, causes $\Omega(2^n)$ iterations in general [26], albeit always finite [33]. A popular workaround is to simply remove Steps 12–14, and such A* without reopening has also been extensively studied [34, 31, 14, 15]. Note that our results are applicable to A* both with and without reopening.

## 4 Lower bounds on the pseudo-dimension

We present lower bounds on the pseudo-dimension for GBFS/A*. We prove the result by constructing $\Omega(n)$ shatterable instances with unweighted graphs. Therefore, the $\mathrm{O}(n \lg n)$ upper bounds for GBFS (Theorem 1) and A* under the edge-weight assumption (Theorem 4) are tight up to a $\lg n$ factor.

**Theorem 5.** *For GBFS/A* $A_\rho$ with parameters $\rho \in \mathbb{R}^n$, it holds that $\mathrm{Pdim}(\mathcal{U}) = \Omega(n)$.*

*Proof.* We construct a series of $n-4$ instances, $x_1, \ldots, x_{n-4}$, that can be shattered by $\mathcal{U}$, where each $u_\rho$ returns the length of an $s$–$t$ path found by $A_\rho$. We label vertices in $V$ by $s$, $r$, $t$, or $i \in [n-3]$. See Figure 1 for an example with $n = 8$. We define $M = V \setminus \{s, r, t\}$. For each $x_i$ ($i \in [n-4]$), we draw edges $(s, v)$ for $v \in M$ and $(v, t)$ for $v \in \{v' \in M \mid v' > i\}$, which constitute optimal $s$–$t$ paths of length 2. In addition, for each $x_i$, we draw edges $(i, r)$ and $(r, t)$, where $s \to i \to r \to t$ is the only suboptimal path of length 3. Letting $t_i = 2.5$ for $i \in [n-4]$, we prove that $\mathcal{U}$ can shatter those $n-4$ instances, i.e., $A_\rho$ can return suboptimal solutions to any subset of $\{x_1 \ldots, x_{n-4}\}$ by appropriately setting $\rho$.

Let $S \subseteq [n-4]$ indicate a subset of instances, to which we will make $A_\rho$ return suboptimal solutions. We show that for any $S$, we can set $\rho$ so that $A_\rho$ returns $s \to i \to r \to t$ to $x_i$ if and only if $i \in S$. We refer to the vertex labeled by $n-3$ as $m$, which we use to ensure that every instance has an optimal path $s \to m \to t$. We set $\rho$ as follows: $\rho_s = n$ (or an arbitrary value), $\rho_r = \rho_t = 0$, $\rho_i = i + 2$ if $i \in S \cup \{m\}$, and $\rho_i = n$ (or a sufficiently large value) if $i \in [n-4] \setminus S$. If $A_\rho$ with this $\rho$ is applied to $x_i$, it iteratively selects vertices in $S \cup \{m\}$ in increasing order of their labels until a vertex that has a child is selected. Once a vertex with a child is expanded, it ends up returning $s \to i \to r \to t$ if $i \in S$ and $s \to v \to t$ for some $v > i$ if $i \notin S$. We below confirm this more precisely, separately for GBFS and A*.

**GBFS.** We consider applying GBFS $A_{\boldsymbol{\rho}}$ to $x_i$. $A_{\boldsymbol{\rho}}$ first expands $s$ and add vertices in $M$ to OPEN. Since vertices in $v \in [n-4] \setminus S$ have sufficiently large scores of $n$, they are never selected before any vertex in $S \cup \{m\}$. Thus, $A_{\boldsymbol{\rho}}$ selects a vertex from $S \cup \{m\}$ with the smallest score. If the selected vertex, denoted by $v$, satisfies $v < i$, there is no child of $v$; hence, nothing is added to OPEN, and we go back to Step 3. In this way, $A_{\boldsymbol{\rho}}$ iteratively moves $v \in S \cup \{m\}$ that has no child from OPEN to CLOSED. Consider the first time when the selected vertex $v \in S \cup \{m\}$ has a child $c$ (this situation is guaranteed to occur since $m$ always has a child). If $i \notin S$, we have $v \neq i$ since $v$ is selected from $S \cup \{m\}$. Then, since $v$'s child is $c = t$, $A_{\boldsymbol{\rho}}$ returns $s \to v \to t$ with $v \neq i$. If $i \in S$, then $i$ has the smallest score ($\rho_i = i + 2$) among all vertices in $S \cup \{m\}$ that have a child. Thus, $A_{\boldsymbol{\rho}}$ selects $i$ and opens $r$. Now, $r$ has the smallest score of $\rho_r = 0$. Therefore, $A_{\boldsymbol{\rho}}$ selects $r$ and reaches $t$, returning $s \to i \to r \to t$. Consequently, $A_{\boldsymbol{\rho}}$ returns the suboptimal path if and only if $i \in S$.

**A\*.** It first expands $s$ and add $M$ to OPEN. Since $g_v = 1$ for all $v \in M$, only $\boldsymbol{\rho}$ values matter when comparing the scores, as with the case of GBFS. Therefore, A\* iterates to move vertices in $S \cup \{m\}$ from OPEN to CLOSED until a vertex that has a child is selected. Consider the first time a selected vertex $v$ has a child $c$ (so far, $s$ is not reopened since $g_s = 0$). As with the case of GBFS, we have $v \neq i$ and $c = t$ if $i \notin S$, or $v = i$ and $c = r$ if $i \in S$. Now, every $v' \in \text{OPEN} \setminus \{c\}$ has a score of at least $4$ since $g_{v'} = 1$ and $\rho_{v'} \geq 3$ for $v' \in M$. Therefore, if $i \notin S$, $t \in \text{OPEN}$ has the smallest score of $g_t + \rho_t = 2 + 0 = 2$. Thus, $A_{\boldsymbol{\rho}}$ next selects $t$ and returns $s \to v \to t$, where $v \neq i$. If $i \in S$, since $r \in \text{OPEN}$ has the smallest score of $g_r + \rho_r = 2 + 0 = 2$, $A_{\boldsymbol{\rho}}$ selects $r$ and opens $t$. Then, since $t$ has the score of $g_t + \rho_t = 3 + 0 = 3$, $A_{\boldsymbol{\rho}}$ selects $t$ and returns $s \to i \to r \to t$. To conclude, $A_{\boldsymbol{\rho}}$ returns the suboptimal path if and only if $i \in S$. $\qquad\square$

## 5 Toward better guarantees on the suboptimality of A\*

Given the results in Sections 3 and 4, a major open problem is to close the $\tilde{O}(n)$ gap[2] between the $O(n^2 \lg n)$ upper bound and the $\Omega(n)$ lower bound for A\* in general cases. This problem seems very complicated, as we will discuss in Section 6. Instead, we here study a particular case where we want to bound the expected suboptimality of A\*, which is an important performance measure since learned heuristic values are not always admissible. We show that a general bound obtained from Theorem 2 can be sometimes improved by using a $\boldsymbol{\rho}$-dependent worst-case bound [34].

For any $x \in \Pi$, let $\text{Opt}(x)$ and $\text{Cost}_{\boldsymbol{\rho}}(x)$ be the costs of an optimal solution and an $s$–$t$ path returned by $A_{\boldsymbol{\rho}}$, respectively, and let $u_{\boldsymbol{\rho}}(x) = \text{Cost}_{\boldsymbol{\rho}}(x) - \text{Opt}(x)$ be the suboptimality. From Theorem 2 and Proposition 1, we can obtain the following high-probability bound on the expected suboptimality:

$$\mathbb{E}_{x \sim \mathcal{D}}[\text{Cost}_{\boldsymbol{\rho}}(x) - \text{Opt}(x)] \leq \frac{1}{N} \sum_{i=1}^{N} (\text{Cost}_{\boldsymbol{\rho}}(x) - \text{Opt}(x)) + \tilde{O}\left(H \sqrt{\frac{n^2 + \lg \frac{1}{\delta}}{N}}\right). \quad (1)$$

That is, the expected suboptimality can be bounded from above by the empirical suboptimality over the $N$ training instances (an empirical term) plus an $\tilde{O}(H\sqrt{n^2/N})$ term (a complexity term). While this bound is useful when $N \gg n^2$, we may not have enough training instances in practice. In such cases, the complexity term becomes dominant and prevents us from obtaining meaningful guarantees. In what follows, we present an alternative bound of the form "an empirical term + a complexity term" that can strike a better balance between the two terms when $N$ is not large enough relative to $n^2$.

To this end, we use the notion of *consistency*. We say $\boldsymbol{\rho}$ is *consistent* if $\rho_v \leq \rho_c + w_{(v,c)}$ holds for all $(v, c) \in E$. If $\boldsymbol{\rho}$ is consistent, A\* without reopening returns an optimal solution. Valenzano et al. [34, Theorem 4.6] revealed that for any instance $x \in \Pi$, the suboptimality of A\* can be bounded by the inconsistency accumulated along an optimal path (excluding the first edge containing $s$) as follows:[3]

$$\text{Cost}_{\boldsymbol{\rho}}(x) - \text{Opt}(x) \leq \Delta_{\boldsymbol{\rho}}(x) \coloneqq \sum_{(v,c) \in S^*(x), v \neq s} \max\{\rho_v - \rho_c - w_{(v,c)}, 0\}, \quad (2)$$

where $S^*(x) \subseteq E$ is an optimal solution to $x$ (if there are multiple optimal solutions, we break ties by using the lexicographical order induced from the total order defined in Assumption 2). We call $\Delta_{\boldsymbol{\rho}}(x)$ the inconsistency (of $\boldsymbol{\rho}$ on $S^*(x)$).

---

[2] We use $\tilde{O}$ and $\tilde{\Omega}$ to hide logarithmic factors of $n$ and $N$ for simplicity.

[3] The original theorem in [34] is applicable only to the case where A\* does not reopen vertices and $\rho_t = 0$. These restrictions are unnecessary as detailed in Appendix C, and thus our result holds regardless of reopening.

Given $N$ instances $x_1, \ldots, x_N$, we can compute the empirical inconsistency, $\frac{1}{N} \sum_{i=1}^{N} \Delta_{\boldsymbol{\rho}}(x_i)$, at the cost of solving the $N$ instances, which we will use as an empirical term. To define the corresponding complexity term, we regard $\Delta_{\boldsymbol{\rho}}(\cdot) : \Pi \to [0, \hat{H}]$ as an inconsistency function parameterized by $\boldsymbol{\rho}$, where we will discuss how large $\hat{H} > 0$ can be later, and we let $\hat{\mathcal{U}} = \{\, \Delta_{\boldsymbol{\rho}} : \Pi \to [0, \hat{H}] \mid \boldsymbol{\rho} \in \mathbb{R}^n \,\}$. The following theorem says that the class $\hat{\mathcal{U}}$ of inconsistency functions has a smaller pseudo-dimension than the class $\mathcal{U}$ of general utility functions.

**Theorem 6.** *For the class $\hat{\mathcal{U}}$ of inconsistency functions, it holds that* $\mathrm{Pdim}(\hat{\mathcal{U}}) = \mathrm{O}(n \lg n)$.

By using (2), Proposition 1, and Theorem 6, we can obtain the following high-probability bound on the expected suboptimality, whose complexity term has a better dependence on $n$ than that of (1):

$$\mathbb{E}_{x \sim \mathcal{D}}[\mathrm{Cost}_{\boldsymbol{\rho}}(x) - \mathrm{Opt}(x)] \leq \mathbb{E}_{x \sim \mathcal{D}}[\Delta_{\boldsymbol{\rho}}(x)] \leq \frac{1}{N} \sum_{i=1}^{N} \Delta_{\boldsymbol{\rho}}(x_i) + \tilde{\mathrm{O}}\left( \hat{H} \sqrt{\frac{n + \lg \frac{1}{\delta}}{N}} \right).$$

This bound is uniform for all $\boldsymbol{\rho} \in \mathbb{R}^n$, as with other bounds discussed so far. Thus, the bound holds even if we choose $\boldsymbol{\rho}$ to minimize the empirical inconsistency. Note that the empirical inconsistency is convex in $\boldsymbol{\rho}$ since $\Delta_{\boldsymbol{\rho}}(x_i)$ consists of a maximum of a linear function of $\boldsymbol{\rho}$ and zero, hence easier to minimize than the raw empirical suboptimality in practice (and suitable for a recent online-convex-optimization framework [25]).

Before proving Theorem 6, we present a typical example to show that the inconsistency is not too large relative to the suboptimality.

**Example 2.** Suppose that every edge weight $w_e$ is bounded to $[0, W]$, which ensures that the suboptimality $u_{\boldsymbol{\rho}}$ is at most $H = W(n-1)$ for any $\boldsymbol{\rho} \in \mathbb{R}^n$. For simplicity, we consider the following natural way to compute $\boldsymbol{\rho}$ values: compute an estimate $\hat{w}_e \in [0, W]$ of $w_e$ for each $e \in E$ and let $\rho_v$ be the cost of a shortest $v$–$t$ path with respect to $\{\hat{w}_e\}_{e \in E}$. Then, $\boldsymbol{\rho}$ enjoys the consistency with respect to $\{\hat{w}_e\}_{e \in E}$, i.e., $\rho_v \leq \rho_c + \hat{w}_{(v,c)}$ for every $(v, c) \in E$. Therefore, it holds that

$$\Delta_{\boldsymbol{\rho}}(x) \leq \sum_{(v,c) \in S^*(x), v \neq s} \max\{\rho_v - \rho_c - w_{(v,c)}, 0\} \leq \sum_{(v,c) \in S^*(x), v \neq s} |\hat{w}_{(v,c)} - w_{(v,c)}|.$$

Hence $\Delta_{\boldsymbol{\rho}}$ is at most $\hat{H} = W(n-2)$, implying that $\Delta_{\boldsymbol{\rho}}$ does not largely exceed the suboptimality. If empirically accurate estimates $\hat{w}_e$ for $e \in S^*(x)$ are available, the inconsistency becomes small.

We prove Theorem 6 by using the general analysis framework by Balcan et al. [8]. To begin with, we introduce Assouad's dual class, which provides a formal definition of the class of functions of $\boldsymbol{\rho}$.

**Definition 2** (Assouad [3]). Given a class, $\mathcal{H} \subseteq \mathbb{R}^{\mathcal{Y}}$, of functions $h : \mathcal{Y} \to \mathbb{R}$, the *dual class* of $\mathcal{H}$ is defined as $\mathcal{H}^* = \{\, h_y^* : \mathcal{H} \to \mathbb{R} \mid y \in \mathcal{Y} \,\}$ such that $h_y^*(h) = h(y)$ for each $y \in \mathcal{Y}$.

In our case, we have $\mathcal{Y} = \Pi$ and $\mathcal{H} = \hat{\mathcal{U}}$, and each $\Delta_x^* \in \hat{\mathcal{U}}^*$ is associated with an instance $x \in \Pi$. The following definition will be used to capture the piecewise structure of the dual class $\hat{\mathcal{U}}^*$.

**Definition 3** (Balcan et al. [8, Definition 3.2]). A class, $\mathcal{H} \subseteq \mathbb{R}^{\mathcal{Y}}$, of functions is $(\mathcal{F}, \mathcal{B}, K)$-*piecewise decomposable* for a class $\mathcal{B} \subseteq \{0, 1\}^{\mathcal{Y}}$ of boundary functions and a class $\mathcal{F} \subseteq \mathbb{R}^{\mathcal{Y}}$ of piece functions if the following condition holds: for every $h \in \mathcal{H}$, there exist $K$ boundary functions $b^{(1)}, \ldots, b^{(K)} \in \mathcal{B}$ and a piece function $f_{\boldsymbol{b}}$ for each binary vector $\boldsymbol{b} \in \{0, 1\}^K$ such that for all $y \in \mathcal{Y}$, it holds that $h(y) = f_{\boldsymbol{b}_y}(y)$ where $\boldsymbol{b}_y = (b^{(1)}(y), \ldots, b^{(K)}(y)) \in \{0, 1\}^K$.

The following result of [8] provides an upper bound on the pseudo-dimension of a class of functions via the piecewise structure of the dual class.

**Proposition 2** (Balcan et al. [8, Theorem 3.3]). *Let $\mathcal{U} \subseteq \mathbb{R}^{\Pi}$ be a class of functions. If $\mathcal{U}^* \subseteq \mathbb{R}^{\mathcal{U}}$ is $(\mathcal{F}, \mathcal{B}, K)$-piecewise decomposable with a class $\mathcal{B} \subseteq \{0, 1\}^{\mathcal{U}}$ of boundary functions and a class $\mathcal{F} \subseteq \mathbb{R}^{\mathcal{U}}$ of piece functions, the pseudo-dimension of $\mathcal{U}$ is bounded as follows:*

$$\mathrm{Pdim}(\mathcal{U}) = \mathrm{O}((\mathrm{Pdim}(\mathcal{F}^*) + \mathrm{VCdim}(\mathcal{B}^*)) \lg(\mathrm{Pdim}(\mathcal{F}^*) + \mathrm{VCdim}(\mathcal{B}^*)) + \mathrm{VCdim}(\mathcal{B}^*) \lg K).$$

Now, we are ready to prove Theorem 6.

*Proof of Theorem 6.* Since there is a one-to-one correspondence between $\Delta_{\boldsymbol{\rho}} \in \hat{\mathcal{U}}$ and $\boldsymbol{\rho} \in \mathbb{R}^n$, we below identify $\Delta_{\boldsymbol{\rho}}$ with $\boldsymbol{\rho}$ for simplicity. Let $\mathcal{B} = \left\{ \mathbb{I}(\langle \boldsymbol{z}, \boldsymbol{\rho} \rangle + z_0) \mid (z_0, \boldsymbol{z}) \in \mathbb{R}^{n+1} \right\} \subseteq \{0,1\}^{\hat{\mathcal{U}}}$ and $\mathcal{F} = \left\{ \langle \boldsymbol{z}, \boldsymbol{\rho} \rangle + z_0 \mid (z_0, \boldsymbol{z}) \in \mathbb{R}^{n+1} \right\} \subseteq \mathbb{R}^{\hat{\mathcal{U}}}$ be classes of boundary and piece functions, respectively. We show that $\hat{\mathcal{U}}^*$ is $(\mathcal{F}, \mathcal{B}, \mathrm{O}(n^2))$-piecewise decomposable.

Fix any $\Delta_x^* \in \hat{\mathcal{U}}^*$; this also uniquely specifies an instance $x \in \Pi$ and an optimal solution $S^*(x) \subseteq E$ (due to the tie-breaking). We consider $K = |E| = \mathrm{O}(n^2)$ boundary functions of form $b^{(v,c)}(\boldsymbol{\rho}) = \mathbb{I}(\rho_v - \rho_c - w_{(v,c)} > 0)$ for all edges $(v,c) \in E$. We below confirm that these boundary functions partition $\mathbb{R}^n \ni \boldsymbol{\rho}$ into some regions so that in each region, $\Delta_x^*(\boldsymbol{\rho})$ can be written as a linear function of $\boldsymbol{\rho}$, which belongs to $\mathcal{F}$. For each binary vector $\boldsymbol{b}_{\boldsymbol{\rho}} = \left( b^{(v,c)}(\boldsymbol{\rho}) \right)_{(v,c) \in E} \in \{0,1\}^K$, we define a subset $S_{\boldsymbol{\rho}}(x)$ of $S^*(x)$ as $S_{\boldsymbol{\rho}}(x) = \left\{ (v,c) \in S^*(x) \mid b^{(v,c)}(\boldsymbol{\rho}) = 1, v \neq s \right\}$; that is, each $(v,c) \in S_{\boldsymbol{\rho}}(x)$ satisfies $v \neq s$ and $\rho_v - \rho_c - w_{(v,c)} > 0$. From the definition of $\Delta_{\boldsymbol{\rho}}(x)$, we have $\Delta_x^*(\boldsymbol{\rho}) = \Delta_{\boldsymbol{\rho}}(x) = \sum_{(v,c) \in S_{\boldsymbol{\rho}}(x)} (\rho_v - \rho_c - w_{(v,c)})$. This is a linear function of $\boldsymbol{\rho}$, and thus we can choose a piece function $f_{\boldsymbol{b}_{\boldsymbol{\rho}}} \in \mathcal{F}$ such that $\Delta_x^* = f_{\boldsymbol{b}_{\boldsymbol{\rho}}}$. This relation holds for every $\boldsymbol{b}_{\boldsymbol{\rho}} \in \{0,1\}^K$, and thus we have $\Delta_x^*(\boldsymbol{\rho}) = f_{\boldsymbol{b}_{\boldsymbol{\rho}}}(\boldsymbol{\rho})$ for all $\boldsymbol{\rho} \in \mathbb{R}^n$. Hence $\hat{\mathcal{U}}^*$ is $(\mathcal{F}, \mathcal{B}, \mathrm{O}(n^2))$-piecewise decomposable.

Since $\mathcal{F}^*$ and $\mathcal{B}^*$ can be seen as classes of linear and halfspace functions of $(z_0, \boldsymbol{z}) \in \mathbb{R}^{n+1}$, respectively, we have $\mathrm{Pdim}(\mathcal{F}^*) = \mathrm{VCdim}(\mathcal{B}^*) = n + 1$ (see also [6, Lemma 3.10], a preprint version of [8]). Therefore, from Proposition 2, we obtain $\mathrm{Pdim}(\hat{\mathcal{U}}) = \mathrm{O}(n \lg n)$. $\qquad\square$

## 6 Conclusion and discussion

We have studied the sample complexity of learning heuristic functions for GBFS and A* on graphs with a fixed vertex set of size $n$. For GBFS and A*, we have proved that the pseudo-dimensions are upper bounded by $\mathrm{O}(n \lg n)$ and $\mathrm{O}(n^2 \lg n)$, respectively. As for A*, we have shown that the bound can be improved to $\mathrm{O}(n^2 \lg d)$ if every vertex has a degree of at most $d$ and to $\mathrm{O}(n \lg n)$ if edge weights are bounded integers. We have also presented the $\Omega(n)$ lower bounds for GBFS and A*, implying that our bounds for GBFS and A* under the integer-weight condition are tight up to a $\lg n$ factor. Finally, we have discussed bounds on the suboptimality of A* and obtained a guarantee with a better complexity term by evaluating the pseudo-dimension of the class of inconsistency functions.

An open problem is to close the gap between the upper and lower bounds regarding A* for general cases. This, however, does not seem straightforward. We here discuss the reasons for the difficulty. As regards the upper bound, the bottleneck is the bound of $(n-2)!$ on $|\mathcal{G}_v|$, but this cannot be improved in general, as shown in Example 1. Considering this, the direct use of Sauer's lemma would not yield better upper bounds. Thus, we need to use some special structures of the hyperplanes (e.g., each has only two variables), which would require more complicated analysis. As for the lower bound, the construction of the $\Omega(n)$ instances in Theorem 5 relies on the fact that $\boldsymbol{\rho}$ has an $n$ degree of freedom. In addition, Theorem 4 implies that we need to consider instances with non-integer edge weights (or exponentially large integer weights in $n$) to obtain a lower bound of $\tilde{\Omega}(n^2)$. Considering the above, we would need more involved techniques for constructing a set of $\tilde{\Omega}(n^2)$ shatterable instances.

Another interesting future direction is to improve upper bounds on the pseudo-dimension by restricting heuristic functions to some classes. Appendix D will present an illustrative example where we can achieve $\mathrm{polylog}(n)$ upper bounds on the pseudo-dimension by assuming that heuristic functions with much fewer tunable parameters than $n$ can be designed in an instance-specific manner.

We finally discuss limitations of our work. As mentioned in Section 2, we require every instance to be defined on (subsets of) a fixed vertex set. Also, our work does not cover the case where heuristic function values can change depending on instance-dependent features. Studying how to overcome these limitations would also constitute interesting future work.

## Acknowledgements

The authors thank Siddharth Prasad for kindly telling a general result of [10] and anonymous reviewers for providing valuable suggestions, particularly on Appendix D. This work was supported by JST ERATO Grant Number JPMJER1903 and JSPS KAKENHI Grant Number JP22K17853.

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
