# Appendix

## A    Comparisons with previous results on tree search

We compare our upper bounds with those of existing results on general tree search [9, 10].

Balcan et al. [9] studied the pseudo-dimension for tree-search algorithms in the following situation; a tree-search algorithm with $d$ configurable parameters builds a search tree of size at most $\kappa$ by iteratively choosing an action from a set of at most $T$ possible actions. In this setting, they obtained an $\mathrm{O}(d\kappa \log T + d \log d)$ upper bound. Balcan et al. [10] removed the dependence on $\kappa$ assuming scores governing the tree search to be defined by *path-wise* functions. Their bound is $\mathrm{O}(d\Delta^2 \log k + d\Delta \log T)$, where $\Delta$ and $k$ are the maximum depth and the number of children, respectively, of search trees. Since $\kappa$ can be exponential in the depth $\Delta$, this is a considerable improvement in this context.

In our setting, since there are $n$ configurable parameters, $\boldsymbol{\rho} \in \mathbb{R}^n$, we have $d = n$. If we apply the bound of [9] to GBFS/A* regarded as a tree-search algorithm that iteratively builds search states, $\kappa$ and $T$ values can be as large as $\Omega(n)$. This is because GBFS/A* can perform $\Omega(n)$ iterations, where each iteration increases the tree size, and the number of possible actions is equal to the size of OPEN, which is $\Omega(n)$ in general. Moreover, for A* with reopening, the number of iterations can be $\Omega(2^n)$ as mentioned in Section 3.2, implying $\kappa = \Omega(2^n)$. Thus, only in the case of A* without reopening, the previous bound matches our $\mathrm{O}(n^2 \lg n)$ bound (Theorem 2). As for GBFS and A* with reopening, our Theorems 1 and 2 provide $\mathrm{O}(n \lg n)$ and $\mathrm{O}(n^2 \lg n)$ bounds, respectively, which improve the $\mathrm{O}(n^2 \lg n)$ and $\mathrm{O}(n2^n \lg n)$ bounds, respectively, implied by the previous result.

As for the result of [10], seeing GBFS/A* as a tree-search algorithm again, the maximum tree depth is as large as the number of vertices in general, i.e., $\Delta = \Omega(n)$. Also, the number of children can be as large as the size of OPEN, hence $k = \Omega(n)$. Thus, the result of [10] leads to an $\mathrm{O}(n^3 \lg n)$ bound, which is larger than any of our upper bounds.

## B    How to deal with varying $t$

As mentioned in Section 2.2, given a fixed $t \in V$, each entry $\rho_v$ in $\boldsymbol{\rho}$ indicates an estimated cost of the shortest $v$–$t$ path. Therefore, if $t$ changes over instances, we need to define $\boldsymbol{\rho}$ for each $t$, which we here denote by $\boldsymbol{\rho}^t \in \mathbb{R}^n$. If $t$ changes, the structure of path-finding instances also greatly changes. Thus, it is natural to evaluate the performance of GBFS/A* separately for each $t$. Specifically, we take $\mathcal{D}$ to be a conditional distribution, from which path-finding instances with fixed $t$ are drawn, and we analyze the sample complexity for each possible $t \in V$. In practice, $\{\boldsymbol{\rho}^t\}_{t \in V}$ may be obtained by, e.g., learning a function that embeds vertices into a metric space and measuring distances on the space, as mentioned in Section 2.2. In this case, an embedding function with tunable parameters governs all the $\{\boldsymbol{\rho}^t\}_{t \in V}$ values. Considering such situations, we need to bound the sample complexity of learning heuristic functions for all possible $t \in V$. This can be done at a slight cost of increasing the bound in Proposition 1 by taking a union bound over all possible $t \in V$. (Note that the upper bounds on the pseudo-dimension in Theorems 1, 2 and 6 hold separately for each $t \in V$ by regarding $\boldsymbol{\rho}$ as $\boldsymbol{\rho}^t$.) Since there are at most $n$ possible choices of $t$, this replaces $\delta$ in Proposition 1 with $\delta/n$, yielding only a $\lg n$ additive factor. This effect is small relative to that of the pseudo-dimension term.

## C    Worst-case analysis of A* regardless of reopening

We show that the existing bound on the suboptimality of A* by Valenzano et al. [34] holds regardless of whether we allow A* to reopen vertices or not, and we also remove a minor assumption of $\rho_t = 0$. Note that the result of [34], which focuses on the case without reopening, does not immediately imply the same bound for A* with reopening since reopening sometimes degrades the solution quality [31].

We fix an instance and define the inconsistency of an edge $(v, v') \in E$ as

$$\mathrm{Inc}(v, v') = \max\{\rho_v - \rho_{v'} - w_{(v,v')}, 0\}. \tag{A1}$$

Fix an optimal $s$–$t$ path and let $P_{\mathrm{opt}} = v_0, v_1, \ldots, v_k$ be a sequence of vertices on the optimal path, where $v_0 = s$, $v_k = t$, and the optimal $s$–$t$ path visits the vertices in this order. Suppose that $v_k = t$ is first selected at the $(T + 1)$st iteration, at which the algorithm terminates.

**Theorem 7.** *Let* Cost *be the cost of an s–t path returned by A\* (Algorithm 2) with/without reopening, and let* Opt *be the cost of* $P_{\mathrm{opt}} = v_0, v_1, \ldots, v_k$. *It holds that*

$$\mathrm{Cost} \leq \mathrm{Opt} + \sum_{j=1}^{k-1} \mathrm{Inc}(v_j, v_{j+1}).$$

The theorem was proved by Valenzano et al. [34] for Algorithm 2 without reopening. Their proof uses the fact that once a vertex is added to CLOSED, it never gets out of CLOSED. If we allow A\* to reopen vertices, the fact is not always true. Therefore, we need to prove the theorem without using the property of CLOSED. To this end, we define lists of selected vertices, which play a similar role to CLOSED in our proof. Formally, for $\tau = 0, 1, \ldots, T + 1$, we define SELECTED$_\tau$ as a list of vertices that have been selected in Step 3 in the first $\tau$ iterations. Note that even with reopening, once a vertex is added to SELECTED$_\tau$, it never gets out of the list.

As in [34], we derive the bound in Theorem 7 by decomposing $P_{\mathrm{opt}}$ into two subpaths, which are defined based on the following *shallowest* vertex.

**Lemma 3.** *We say a vertex* $v_i \in P_{\mathrm{opt}} = v_0, v_1, \ldots, v_k$ *is the shallowest vertex at* $\tau \in \{0, 1, \ldots, T\}$ *if it satisfies the following conditions after the $\tau$th iteration:*

$$v_i \in \mathtt{OPEN} \setminus \mathtt{SELECTED}_\tau \qquad and \qquad \{v_j \in P_{\mathrm{opt}} \mid j < i\} \subseteq \mathtt{SELECTED}_\tau.$$

*For every* $\tau = 0, 1, \ldots, T$, *a shallowest vertex always exists.*

*Proof.* We prove the claim by induction. If $\tau = 0$, $v_0$ is the shallowest since we have SELECTED$_0 = \emptyset$ and OPEN $= \{v_0\}$. If $\tau = 1$, $v_1$ is the shallowest vertex since we have SELECTED$_1 = \{v_0\}$ and $v_1 \in \mathtt{OPEN} \setminus \mathtt{SELECTED}_1$. Assume that the claim is true for $\tau' < \tau$ and let $v_{i'} \in \mathtt{OPEN} \setminus \mathtt{SELECTED}_{\tau-1}$ be the shallowest vertex at $\tau - 1$. We consider two cases: $v_{i'}$ or $v \neq v_{i'}$ is selected at the $\tau$th iteration. If $v \neq v_{i'}$ is selected, we have SELECTED$_\tau = \mathtt{SELECTED}_{\tau-1} \cup \{v\}$ and $v_{i'} \in \mathtt{OPEN} \setminus \mathtt{SELECTED}_\tau$; thus $v_{i'}$ remains the shallowest at $\tau$. If $v_{i'}$ is selected, take the longest subpath of $P_{\mathrm{opt}}$ that starts from $v_{i'}$ and is contained in SELECTED$_\tau$. We denote such a subpath by $v_{i'}, \ldots, v_{i''}$, where $i'' < k$ holds since $v_k$ is never selected in the first $T$ iterations. From the definition of the subpath, we have $v_{i''+1} \notin \mathtt{SELECTED}_\tau$. Moreover, $v_{i''+1}$ must have been opened since its parent $v_{i''}$ has been selected. Thus, $v_{i''+1} \in \mathtt{OPEN} \setminus \mathtt{SELECTED}_\tau$ holds. Furthermore, we have $\{v_0, \ldots, v_{i'-1}\} \subseteq \mathtt{SELECTED}_\tau$ due to the induction hypothesis and $\{v_{i'}, \ldots, v_{i''}\} \subseteq \mathtt{SELECTED}_\tau$ from the definition of the subpath. Thus, $v_{i''+1}$ is the shallowest vertex at $\tau$. To conclude, the shallowest vertex at $\tau$ exists in any case. The proof is completed by induction. □

### C.1 Decomposing the suboptimality term with the shallowest vertex

For every $v \in V$, we let $g_v^*$ and $\rho_v^*$ denote the costs of optimal $s$–$v$ and $v$–$t$ paths, respectively. We use $g_{v,v'}^*$ to denote the cost of an optimal $v$–$v'$ path for any pair of $v, v' \in V$; it holds that $g_v^* = g_{s,v}^*$. For each $v \in V$, let $g_v^{(\tau)}$ be the $g_v$ value after the $\tau$th iteration, where $g_s^{(0)} = 0$ and $g_v^{(0)} = \infty$ for $v \neq s$. If $g_v$ is updated in the $(\tau+1)$st iteration, we have $g_v^{(\tau+1)} < g_v^{(\tau)}$; otherwise we have $g_v^{(\tau+1)} = g_v^{(\tau)}$. Thus, $g_v^{(\tau)}$ is non-increasing in $\tau$. We define $\delta g_v^{(\tau)} = g_v^{(\tau)} - g_v^*$ as the $g$-cost error of $v$ after the $\tau$th iteration, which is also non-increasing in $\tau$.

The following lemma states that the suboptimality can be decomposed into two parts: a $g$-cost error of the shallowest vertex $v_i$ and the inadmissibility of $\rho_{v_i}$ (subtracted by $\rho_t$).

**Lemma 4.** *If* $v_i$ *is the shallowest vertex at $T$, it holds that*

$$\mathrm{Cost} \leq \mathrm{Opt} + \delta g_{v_i}^{(T)} + \rho_{v_i} - \rho_{v_i}^* - \rho_t.$$

*Proof.* After the $T$th iteration, the score of $v_i$ is

$$\begin{aligned}
g_{v_i}^{(T)} + \rho_{v_i} &= g_{v_i}^* + \delta g_{v_i}^{(T)} + \rho_{v_i} \\
&= g_{v_i}^* + \rho_{v_i}^* + \delta g_{v_i}^{(T)} + \rho_{v_i} - \rho_{v_i}^* \\
&= \mathrm{Opt} + \delta g_{v_i}^{(T)} + \rho_{v_i} - \rho_{v_i}^*.
\end{aligned}$$

Since $v_k = t$ is selected at the $(T+1)$st iteration instead of $v_i$, it holds that $g_t^{(T)} + \rho_t \leq g_{v_i}^{(T)} + \rho_{v_i}$. Since we have Cost $\leq g_t^{(T)}$, we obtain the statement by rearranging the terms. □

## C.2 Bounding $\rho_{v_i} - \rho_{v_i}^* - \rho_t$

We prove a general lemma for later use, which implies an upper bound on $\rho_{v_i} - \rho_{v_i}^* - \rho_t$.

**Lemma 5.** *Let $P = v_0, v_1, \ldots, v_k$ be any optimal $v_0$–$v_k$ path. It holds that*

$$\rho_{v_0} - \rho_{v_k} - g_{v_0,v_k}^* \leq \sum_{i=0}^{k-1} \mathrm{Inc}(v_i, v_{i+1}).$$

*Proof.* From the definition (A1), $\mathrm{Inc}(v_i, v_{i+1}) \geq \rho_{v_i} - \rho_{v_{i+1}} - w_{(v_i,v_{i+1})}$ holds. Therefore, we have

$$\sum_{i=0}^{k-1} \left( \rho_{v_i} - \rho_{v_{i+1}} - w_{(v_i,v_{i+1})} \right) \leq \sum_{i=0}^{k-1} \mathrm{Inc}(v_i, v_{i+1}).$$

Using a telescoping sum argument, we obtain

$$\rho_{v_0} - \rho_{v_k} - \sum_{i=0}^{k-1} w_{(v_i,v_{i+1})} \leq \sum_{i=0}^{k-1} \mathrm{Inc}(v_i, v_{i+1}).$$

Since $P$ is optimal, we have $\sum_{i=0}^{k-1} w_{(v_i,v_{i+1})} = g_{v_0,v_k}^*$, thus completing the proof. $\qquad\square$

Consider applying Lemma 5 to the subpath $P = v_i, \ldots, v_k$ of $P_{\mathrm{opt}}$, which is an optimal $v_i$–$v_k$ path. Since $g_{v_i,v_k}^* = \rho_{v_i}^*$ and $\rho_{v_k} = \rho_t$, it holds that

$$\rho_{v_i} - \rho_{v_i}^* - \rho_t \leq \sum_{j=i}^{k-1} \mathrm{Inc}(v_j, v_{j+1}). \tag{A2}$$

Thus, we can obtain an upper bound on $\rho_{v_i} - \rho_{v_i}^* - \rho_t$.

## C.3 Bounding $\delta g_{v_i}^{(T)}$

Our goal is to prove the following lemma.

**Lemma 6.** *Let $P_{\mathrm{opt}} = v_0, \ldots, v_k$ be the optimal $s$–$t$ path in the statement of Theorem 7. Then, the shallowest vertex $v_i \in P_{\mathrm{opt}}$ at $T$ satisfies*

$$\delta g_{v_i}^{(T)} \leq \sum_{j=1}^{i-1} \mathrm{Inc}(v_j, v_{j+1}).$$

To prove Lemma 6, we need the following two lemmas.

**Lemma 7.** *For $P_{\mathrm{opt}}$ in Lemma 6 and $i \geq 1$, if $v_{i-1} \in P_{\mathrm{opt}}$ is first selected at the $\tau'$th iteration and satisfies $\delta g_{v_{i-1}}^{(\tau')} \leq \sum_{j=1}^{i-2} \mathrm{Inc}(v_j, v_{j+1})$, then $v_i \in P_{\mathrm{opt}}$ satisfies $\delta g_{v_i}^{(\tau)} \leq \sum_{j=1}^{i-1} \mathrm{Inc}(v_j, v_{j+1})$ for all $\tau = \tau', \ldots, T$.*

*Proof.* In the $\tau'$th iteration, we update $g_{v_i}$ if it is larger than $g_{\mathrm{new}} = g_{v_{i-1}}^{(\tau')} + w_{(v_{i-1},v_i)}$, hence $g_{v_i}^{(\tau')} \leq g_{v_{i-1}}^{(\tau')} + w_{(v_{i-1},v_i)}$. Since $(v_{i-1}, v_i)$ is an edge on $P_{\mathrm{opt}}$, $g_{v_i}^* = g_{v_{i-1}}^* + w_{(v_{i-1},v_i)}$ holds. Therefore, it holds that $\delta g_{v_i}^{(\tau')} \leq \delta g_{v_{i-1}}^{(\tau')}$. Since $\delta g_{v_i}^{(\tau)}$ is non-increasing in $\tau$, we have $\delta g_{v_i}^{(\tau)} \leq \delta g_{v_i}^{(\tau')}$. If $i = 1$, since $v_0 = s$, we obtain

$$\delta g_{v_1}^{(\tau)} \leq \delta g_{v_1}^{(\tau')} \leq \delta g_{v_0}^{(\tau')} = g_{v_0}^{(\tau')} - g_{v_0}^* = 0 - 0 = 0.$$

If $i > 1$, we have

$$\delta g_{v_i}^{(\tau)} \leq \delta g_{v_i}^{(\tau')} \leq \delta g_{v_{i-1}}^{(\tau')} \leq \sum_{j=1}^{i-2} \mathrm{Inc}(v_j, v_{j+1}) \leq \sum_{j=1}^{i-1} \mathrm{Inc}(v_j, v_{j+1}),$$

where we used the assumption on $\delta g_{v_{i-1}}^{(\tau')}$ and $\mathrm{Inc}(v_{i-1}, v_i) \geq 0$. $\qquad\square$

**Lemma 8.** *For $P_{\mathrm{opt}}$ in Lemma 6 and any $\tau = 0, 1, \ldots, T$, every $v_i \in P_{\mathrm{opt}} \cap \mathtt{SELECTED}_\tau$ satisfies* $\delta g_{v_i}^{(\tau)} \leq \sum_{j=1}^{i-1} \mathrm{Inc}(v_j, v_{j+1})$.

*Proof.* The proof is by induction on $\tau$. If $\tau = 0$, the claim is vacuously true since $\mathtt{SELECTED}_\tau = \emptyset$. If $\tau = 1$, only $v_0$ is in $\mathtt{SELECTED}_1$. Since $g_{v_0}^{(1)} = g_{v_0}^* = 0$, we have $\delta g_{v_0}^{(1)} = 0$. Thus, the claim is true.

Assume that the claim is true after the first $\tau \geq 1$ iterations; in other words, for any $\tau' \leq \tau$, every $v_{i'} \in P_{\mathrm{opt}} \cap \mathtt{SELECTED}_{\tau'}$ satisfies $\delta g_{v_{i'}}^{(\tau')} \leq \sum_{j=1}^{i'-1} \mathrm{Inc}(v_j, v_{j+1})$. Since $\delta g_{v_{i'}}^{(\tau)}$ is non-increasing in $\tau$, from the induction hypothesis, vertices in $P_{\mathrm{opt}} \cap \mathtt{SELECTED}_\tau$ remain to satisfy the inequality after the $(\tau + 1)$st iteration. Therefore, we focus on the vertex selected at the $(\tau + 1)$st iteration, which is the only new vertex in $\mathtt{SELECTED}_{\tau+1}$. If the selected vertex is not in $P_{\mathrm{opt}}$, the statement is true after the $(\tau + 1)$st iteration. We below discuss the case where the selected vertex is in $P_{\mathrm{opt}}$. We let $v_i \in P_{\mathrm{opt}}$ be the selected vertex, where $i \geq 1$, and discuss two cases: $v_i$'s parent, $v_{i-1}$, is in $\mathtt{SELECTED}_\tau$ or not.

**Case 1:** $v_{i-1} \in \mathtt{SELECTED}_\tau$. Suppose that $v_{i-1}$ has been first selected at the $\tau'$th iteration ($\tau' \leq \tau$). The induction hypothesis implies $\delta g_{v_{i-1}}^{(\tau')} \leq \sum_{j=1}^{i-2} \mathrm{Inc}(v_j, v_{j+1})$. Thus, from Lemma 7, we obtain $\delta g_{v_i}^{(\tau+1)} \leq \sum_{j=1}^{i-1} \mathrm{Inc}(v_j, v_{j+1})$.

**Case 2:** $v_{i-1} \notin \mathtt{SELECTED}_\tau$. In this case, we have $i > 1$ since $v_0$ is selected at the first iteration. Let $v_{i'}$ be the shallowest vertex at $\tau$. Since $v_j \in \mathtt{SELECTED}_\tau$ must hold for all $j < i'$, we have $i' < i$. Since $v_i$ is selected instead of $v_{i'}$, it holds that $g_{v_i}^{(\tau)} + \rho_{v_i} \leq g_{v_{i'}}^{(\tau)} + \rho_{v_{i'}}$. Therefore, we have

$$\delta g_{v_i}^{(\tau)} \leq g_{v_{i'}}^{(\tau)} - g_{v_i}^* + \rho_{v_{i'}} - \rho_{v_i} = \delta g_{v_{i'}}^{(\tau)} + g_{v_{i'}}^* - g_{v_i}^* + \rho_{v_{i'}} - \rho_{v_i} = \delta g_{v_{i'}}^{(\tau)} + \rho_{v_{i'}} - \rho_{v_i} - g_{v_{i'},v_i}^*.$$

We below consider bounding the right-hand side. First, we discuss a bound on $\delta g_{v_{i'}}^{(\tau)}$. Suppose that $v_{i'-1} \in \mathtt{SELECTED}_\tau$ is first selected at the $\tau'$th iteration, where $\tau' \leq \tau$. From the induction hypothesis, we have $\delta g_{v_{i'-1}}^{(\tau')} \leq \sum_{j=1}^{i'-2} \mathrm{Inc}(v_j, v_{j+1})$. Therefore, Lemma 7 implies $\delta g_{v_{i'}}^{(\tau)} \leq \sum_{j=1}^{i'-1} \mathrm{Inc}(v_j, v_{j+1})$. Next, from Lemma 5, we have $\rho_{v_{i'}} - \rho_{v_i} - g_{v_{i'},v_i}^* \leq \sum_{j=i'}^{i-1} \mathrm{Inc}(v_j, v_{j+1})$. Consequently, we obtain

$$\delta g_{v_i}^{(\tau+1)} \leq \delta g_{v_i}^{(\tau)} \leq \sum_{j=1}^{i'-1} \mathrm{Inc}(v_j, v_{j+1}) + \sum_{j=i'}^{i-1} \mathrm{Inc}(v_j, v_{j+1}) = \sum_{j=1}^{i-1} \mathrm{Inc}(v_j, v_{j+1}).$$

To conclude, every $v_i \in P_{\mathrm{opt}} \cap \mathtt{SELECTED}_{\tau+1}$ satisfies $\delta g_{v_i}^{(\tau+1)} \leq \sum_{j=1}^{i-1} \mathrm{Inc}(v_j, v_{j+1})$. Therefore, the statement is true by induction. $\qquad\square$

Now, we are ready to prove Lemma 6.

*Proof of Lemma 6.* Since $v_i$ is the shallowest at $T$, $v_{i-1}$ has been first selected at some $\tau$th iteration, where $\tau \leq T$, i.e., $v_{i-1} \in P_{\mathrm{opt}} \cap \mathtt{SELECTED}_\tau$. Thus, Lemma 8 implies $\delta g_{v_{i-1}}^{(\tau)} \leq \sum_{j=1}^{i-2} \mathrm{Inc}(v_j, v_{j+1})$. Therefore, from Lemma 7, we obtain $\delta g_{v_i}^{(T)} \leq \sum_{j=1}^{i-1} \mathrm{Inc}(v_j, v_{j+1})$. $\qquad\square$

### C.4 Proof of Theorem 7

By summing up the above lemmas and equations, we prove Theorem 7.

*Proof of Theorem 7.* By using Lemmas 4 and 6 and eq. (A2), we obtain

$$\begin{aligned}
\mathrm{Cost} &\leq \mathrm{Opt} + \delta g_{v_i}^{(T)} + \rho_{v_i} - \rho_{v_i}^* - \rho_t \\
&\leq \mathrm{Opt} + \sum_{j=1}^{i-1} \mathrm{Inc}(v_j, v_{j+1}) + \sum_{j=i}^{k-1} \mathrm{Inc}(v_j, v_{j+1}) \\
&= \mathrm{Opt} + \sum_{j=1}^{k-1} \mathrm{Inc}(v_j, v_{j+1}). \qquad\qquad\square
\end{aligned}$$

# D An example of improving upper bounds with instance-specific heuristics

GBFS/A* is often applied to path-finding instances that have extremely many vertices, for which our bounds on the pseudo-dimension depending on $n$ or $n^2$ imply somewhat pessimistic sample complexity bounds. To exhibit more practical results for such cases, we study an example situation where we can exponentially improve the upper bounds by using instance-specific heuristic functions.

We assume the vertex set $V$ to be fixed as in Assumption 1 and that each vertex $v \in V$ corresponds to a *state* represented by a vector $\boldsymbol{q}_v \in \Sigma^L$, where $\Sigma$ is a finite set of cardinality $B$. For example, if $\Sigma = \{0, 1\}$, we have $B = 2$ and each $v \in V$ corresponds to a state represented by a bit vector of length $L$. Such situations arise, e.g., when applying GBFS/A* to planning instances represented by the STRIPS model [19]. As for heuristic functions, we assume that each $\rho_v$ value is computed as $\rho_v = \boldsymbol{q}_v^\top \boldsymbol{\theta} + \eta$, where $(\boldsymbol{\theta}, \eta) \in \mathbb{R}^{L+1}$ is a vector of tunable parameters. (Although $\eta$ is not essential in the following analysis, we can use it to make every $\rho_v$ non-negative in practice.) Here, an important observation is that the number of tunable parameters is $L + 1$, while $V$ has up to $n = B^L$ vertices. Using this exponential decrease in the number of tunable parameters, we below obtain $\mathrm{poly}(L \lg B) \simeq \mathrm{polylog}(n)$ upper bounds on the pseudo-dimension for GBFS and A* under the integer-weight assumption. That is, while the number of vertices is exponential in $L$, the upper bounds can scale polynomially with $L$.

**GBFS.** As discussed in the proof of Lemma 1, if two heuristic function values $\boldsymbol{\rho}, \boldsymbol{\rho}' \in \mathbb{R}^n$ have the same total order on $V$, it holds $u_{\boldsymbol{\rho}}(x) = u_{\boldsymbol{\rho}'}(x)$ for all instances $x \in \Pi$. Also, note that the total order is uniquely determined by comparing $\rho_v$ and $\rho_{v'}$ for all pairs of $v, v' \in V$. Therefore, if we partition the space $\mathbb{R}^{L+1}$ of tunable parameters into some regions $\mathcal{P}_1, \mathcal{P}_2, \ldots \subseteq \mathbb{R}^{L+1}$ by $\binom{n}{2}$ hyperplanes of form

$$\rho_v = \rho_{v'} \iff \boldsymbol{q}_v^\top \boldsymbol{\theta} + \eta = \boldsymbol{q}_{v'}^\top \boldsymbol{\theta} + \eta \iff (\boldsymbol{q}_v - \boldsymbol{q}_{v'})^\top \boldsymbol{\theta} = 0,$$

then all $(\boldsymbol{\theta}, \eta)$ values belonging to the same region $\mathcal{P}_i$ result in the same $u_{\boldsymbol{\rho}}(x)$ value for all $x \in \Pi$. Hence, when $(\boldsymbol{\theta}, \eta)$ is allowed to take any value in $\mathbb{R}^{L+1}$, the number of distinct tuples of form $(u_{\boldsymbol{\rho}}(x_1), \ldots, u_{\boldsymbol{\rho}}(x_N))$ is bounded by the number of the regions $\mathcal{P}_1, \mathcal{P}_2, \ldots$. As in the proof of Theorem 2, the number of such regions is $\mathrm{O}\left(\left(e\binom{n}{2}\right)^{L+1}\right)$ due to Sauer's lemma. On the other hand, to shatter $N$ instances $x_1, \ldots, x_N$, we need to make $2^N$ distinct tuples of form $(u_{\boldsymbol{\rho}}(x_1), \ldots, u_{\boldsymbol{\rho}}(x_N))$ by varying $(\boldsymbol{\theta}, \eta) \in \mathbb{R}^{L+1}$. Since $n = \mathrm{O}(B^L)$, solving $\mathrm{O}\left(\left(e\binom{n}{2}\right)^{L+1}\right) \geq 2^N$ for the largest $N$ yields an $\mathrm{O}(L \lg n) \simeq \mathrm{O}(L^2 \lg B)$ bound on $\mathrm{Pdim}(\mathcal{U})$.

**A* under the integer-weight assumption.** We can also obtain a $\mathrm{polylog}(n)$ upper bound for A* under the condition of Theorem 4, i.e., all edge weights take non-negative integer values at most $W$. The proof begins with the same discussion as that of Theorem 2. For a fixed instance $x_k \in \Pi$, the number of possible $u_{\boldsymbol{\rho}}(x_k)$ values is at most the number of regions created by hyperplanes of form $g_v(x_k) + \rho_v = g_{v'}(x_k) + \rho_{v'}$ for all pairs of $v, v' \in V$. Since $g_v(x_k)$ can take $nW$ distinct values, the number of those hyperplanes is $\binom{n^2 W}{2}$, as discussed in the proof of Theorem 4. Therefore, given $N$ instances $x_1, \ldots, x_N$, the number of distinct tuples of form $(u_{\boldsymbol{\rho}}(x_1), \ldots, u_{\boldsymbol{\rho}}(x_N))$ is bounded by the number of regions created by $N\binom{n^2 W}{2}$ hyperplanes. Next, as with the above GBFS case, we bound the number of those regions using Sauer's lemma. With the tunable parameters $(\boldsymbol{\theta}, \eta) \in \mathbb{R}^{L+1}$, each hyperplane can be written as

$$g_v(x_k) + \rho_v = g_{v'}(x_k) + \rho_{v'} \iff (\boldsymbol{q}_v - \boldsymbol{q}_{v'})^\top \boldsymbol{\theta} + g_v(x_k) - g_{v'}(x_k) = 0.$$

Sauer's lemma implies that $N\binom{n^2 W}{2}$ such hyperplanes partition $\mathbb{R}^{L+1}$ into $\mathrm{O}\left(\left(eN\binom{n^2 W}{2}\right)^{L+1}\right)$ regions. To shatter the $N$ instances, the number of the regions must be at least $2^N$. Since $n = \mathrm{O}(B^L)$, solving $\mathrm{O}\left(\left(eN\binom{n^2 W}{2}\right)^{L+1}\right) \geq 2^N$ for the largest $N$ yields an $\mathrm{O}(L \lg nW) \simeq \mathrm{O}(L^2 \lg B + L \lg W)$ upper bound on $\mathrm{Pdim}(\mathcal{U})$.

As regards A* for general cases, it is open whether a similar $\mathrm{polylog}(n)$ upper bound can be achieved. The obstacle is the additional $n$ factor, which remains even if the number of tunable parameters decreases. Thus, this problem would be as difficult as closing the gap between the $\Omega(n)$ lower bound and $\tilde{\mathrm{O}}(n^2)$ upper bound.