# OpenReview forum: "Sample Complexity of Learning Heuristic Functions for Greedy-Best-First and A* Search"
_NeurIPS.cc/2022/Conference — NeurIPS 2022 Accept_

### Official Review · Reviewer_zGFs · 2022-07-11

**Rating:** 8
**Confidence:** 2
**Soundness:** 4 excellent
**Presentation:** 3 good
**Contribution:** 3 good

**Summary:**

This theoretical paper presents sample complexity bounds for learning
heuristics for A* and best-first search. It shows an O(nlogn) upper
bound on the pseudo dimension of BFS and O(n^2logn) for A*, with
Omega(n) lower bounds for both. It shows that the upper bounds are
nearly tight, but can be improved for A* when bounding edge weights
and variable degrees. Moreover, when learning a potentially suboptimal
heuristic function, the paper gives an upper bound on the
suboptimality.


**Questions:**

None

**Limitations:**

No direct societal impact.

**Strengths And Weaknesses:**

The paper is relatively straightforward, in the sense that it gives
clear questions and clear answers. It is well written, and explains
the weaknesses of the results, namely the relatively big gap between
the bounds on the pseudodimension of A*, as well as give some
explanation why it is hard to bridge them.

I don't see any major weaknesses. I would suggest that another
interesting direction here is looking at A* for planning. The graph is
obviously exponentially large, so the bounds here are useless, but it
has a compact representation (e.g. the STRIPS model). Could some
heuristics be learned efficiently in that setting?

----------

Typos, etc:

Defn: you use t_1, ..., t_N for the values in the text and z_1...z_N
in the formula

107: disrtibution

154: gaurantees

---

> ### Author Response · Authors · 2022-08-01
> **Response to Reviewer zGFs**
>
> We appreciate the reviewer's careful reading and insightful comments. We respond to the following comment.
>
> > I don't see any major weaknesses. I would suggest that another interesting direction here is looking at A* for planning. The graph is obviously exponentially large, so the bounds here are useless, but it has a compact representation (e.g. the STRIPS model). Could some heuristics be learned efficiently in that setting?
>
> Thank you for the constructive suggestion. As the reviewer mentioned, some path-finding instances have extremely many vertices but have compact representations. Such representations are sometimes helpful in reducing the sample complexity. Below is an illustrative example.
>
> &nbsp;
> ### An example of improving the upper bound with special heuristics
> Suppose that a graph with $n$ vertices is given as a STRIPS model, where each vertex $v \in V$ corresponds to a state represented by a binary vector $\boldsymbol{q}\_v \in \\{0, 1\\}^\ell$. Then, we have $n = 2^\ell$ vertices, i.e., there are exponentially many vertices in $\ell$. As an example of simple heuristic functions, we assume that a heuristic function value $\rho\_v$ for each $v \in V$ is given as $\rho\_v = \boldsymbol{q}\_v^\top \boldsymbol{\theta}$, where $\boldsymbol{\theta} \in \mathbb{R}^\ell$ is a vector of $\ell$ learnable parameters. If we apply GBFS to the graph, a similar discussion to the proof of Theorem 1 implies that the behavior of GBFS is determined by a total order on $n$ values $\\{ \boldsymbol{q}\_v^\top \boldsymbol{\theta} \\}\_{v \in V}$. Such a total order is unique for all $\boldsymbol{\theta} \in \mathbb{R}^\ell$ in an identical region whose boundaries are given by up to $\binom{n}{2}$ hyperplanes of form $(\boldsymbol{q}\_v - \boldsymbol{q}_{v'})^\top \boldsymbol{\theta} = 0$ for $v, v' \in V$. Thus, Sauer's lemma implies that given any $N$ instances, there are at most $\left( \mathrm{e}\binom{n}{2} N \right)^\ell$ such regions. To shatter the $N$ instances, we need $\left( \mathrm{e}\binom{n}{2} N \right)^\ell = \Omega(2^N)$, implying an $\mathrm{O}(\ell \lg n) \simeq \mathrm{O}(\ell^2)$ upper bound on the pseudo-dimension. That is, even though there are $n = 2^\ell$ vertices, the upper bound depends only polynomially on $\ell$. A similar result is true for A* if edge weights are integer, but A* for general weights does not enjoy such a $\mathrm{poly}(\ell)$ bound as it incurs an additional $\mathrm{O}(n)$ factor.
>
> &nbsp;
>
> The above example implies that we can sometimes reduce the sample complexity by using compact representations of graphs to design appropriate heuristic functions with fewer parameters. Studying how to design such appropriate heuristic functions would be an interesting and important future direction. We will elaborate more on this point in the final version.

---

> > ### Comment · Reviewer_zGFs · 2022-08-09
> > **Useful insight**
> >
> > Thank you for your answer. I encourage you to include a version of this in the final version, if accepted.

---

### Official Review · Reviewer_cPFZ · 2022-07-11

**Rating:** 7
**Confidence:** 3
**Soundness:** 3 good
**Presentation:** 3 good
**Contribution:** 3 good

**Summary:**

The paper presents bounds on the sample complexity required for learning heuristic functions to guide greedy best-first search and A* search for solving the shortest path problem in a given graph. The classical approach to best-first search (and heuristic search in general) is to provide it with a handcrafted heuristic (which is typically obtained by solving a relaxed version of the original problem) in order to guide it more effectively towards the optimal solution. However, more recent work aims to learn the guiding heuristic directly from some training data which could be more appealing in some cases. Therefore, deriving bounds on how much data is required to learn a heuristic function with certain guarantees is called for.


**Questions:**

1. What are the practical implications of the proposed bounds? Many interesting domains for GBFS/A* search have exponentially large implicit search spaces (e.g., planning, probabilistic inference, games, etc.) which means that 'n' in this case could be extremely large and therefore one would require an exponentially large training dataset.

[Post Rebuttal] Thanks for your answers. They have clarified my concerns.

**Ethics Review Area:**

["I don’t know"]

**Limitations:**

see above

**Strengths And Weaknesses:**

The paper is fairly well written and organised. The quality of the presentation is overall very good and therefore the paper is relatively easy to follow. Most of the concepts and technical details are introduced and discussed if a fairly clear manner.

I think the paper needs a more detailed running example. Otherwise it's not very easy to follow the details especially for a reader who's not very familiar with this research area.

Minor comments:

- Definition 1: there is a typo, h(y_i) \geq t_i instead of h(y_i) \geq z_i

---

> ### Author Response · Authors · 2022-08-01
> **Response to Reviewer cPFZ**
>
> We sincerely thank the reviewer for providing valuable comments. We answer the following question.
>
> > What are the practical implications of the proposed bounds? Many interesting domains for GBFS/A\* search have exponentially large implicit search spaces (e.g., planning, probabilistic inference, games, etc.) which means that 'n' in this case could be extremely large and therefore one would require an exponentially large training dataset.
>
> As the reviewer mentioned, our result is sometimes pessimistic since $n$ can be extremely large (although it is tight for GBFS and A* for integer edge-weight graphs due to the lower bound). If we encounter such huge graphs in practice, what we should consider next is how to design simple heuristic functions to reduce the sample complexity. Roughly speaking, our bounds depend on $n = |V|$ mainly because $n$ heuristic function values $\rho_v$ for all $v \in V$ are independently learnable. Therefore, using simple heuristic functions with fewer learnable parameters than $n$ would be an effective workaround when $n$ is huge. For an example of such simple heuristic functions, please see the [response to Reviewer zGFs](https://openreview.net/forum?id=FurHLDnmC5v&noteId=87Oj0VKH2dc).
>
> Further discussion on how to design such heuristic functions would go beyond the scope of our work since it requires more instance-specific analysis; therefore, we have left it for future work. Our result on the fundamental case where all $\rho_v$ ($v \in V$) are learnable will be an important first step to understanding the theoretical aspect of learning-based heuristic search, such as [9, 11, 25, 29, 33].

---

### Official Review · Reviewer_YN2Q · 2022-07-12

**Rating:** 7
**Confidence:** 1
**Soundness:** 3 good
**Presentation:** 3 good
**Contribution:** 3 good

**Summary:**

This paper studies the sample complexity for learning heuristic functions for GBFS/A* search on a graph with a fixed number of nodes n. The analysis uses PAC learning framework, and the main results show the upper and lower bound of pseudo dimensions of a class of utility functions in which each utility function associates a search task to a scalar value between 0 and H. The paper also continues to provide upper bounds on the expectation of gaps between the optimal costs and the suboptimal costs, where the expectation was taken over the search task sampled from some distribution D, and the bounds are given in terms of the number of samples and the number of nodes.


**Questions:**

I don't know very well about PAC analysis, but following the steps in the paper, the claims all make sense.
Although the assumption is simplified to a very limited case, where the graph has the same terminal node
and the sampling procedure ends up visiting roughly all the nodes in the graph (n and N in the equations),
the upper and lower bounds may characterize the theoretical limits for sample efficiency for learning heuristics.

How to use these bounds when we learn heuristics for solving graph search problems?
Can you provide a concrete example with the theoretical bounds?


**Limitations:**

I think this work is not relevant to this section.

**Strengths And Weaknesses:**

Strengths are mathematical analysis of the sample complexity for learning heuristic functions for graph search tasks using GBFS/A*.

Weaknesses are that this analysis emphasizes theoretical aspects and missing practical implications of the upper bounds.

---

> ### Author Response · Authors · 2022-08-01
> **Response to Reviewer YN2Q**
>
> We are grateful to the reviewer for providing thoughtful comments. We answer the following question.
>
> > I don't know very well about PAC analysis, but following the steps in the paper, the claims all make sense. Although the assumption is simplified to a very limited case, where the graph has the same terminal node and the sampling procedure ends up visiting roughly all the nodes in the graph (n and N in the equations), the upper and lower bounds may characterize the theoretical limits for sample efficiency for learning heuristics.
> >
> > How to use these bounds when we learn heuristics for solving graph search problems? Can you provide a concrete example with the theoretical bounds?
>
> We can use our PAC bounds similarly to those in statistical learning theory. In words, PAC bounds guarantee that the expected performance on future instances becomes closer to the empirical performance observed on $N$ sampled instances as $N$ grows.
>
> For example, suppose that path-finding instances on graphs with $n$ vertices are drawn from an unknown distribution. We consider accelerating GBFS/A* applied to those instances by learning good heuristic functions. As is often the case in practice, GBFS/A* with empirically good heuristics runs very fast. However, this empirical observation alone provides no theoretical guarantee on how fast it can be on future instances drawn from the unknown distribution (particularly because learning of heuristic functions may result in overfitting to sampled instances).
>
> In such situations, we can use our PAC bounds to guarantee the expected running time on future instances. Roughly speaking, with high probability, the expected running time on future instances can be bounded by the observed empirical running time plus about $\mathrm{O}\left(\sqrt{\frac{\mathrm{poly}(n)}{N}}  \right)$, where $\mathrm{poly}(n) = n$ for GBFS and $n^2$ for A*. Thus, if we are given about $N \gtrsim \frac{\mathrm{poly}(n)}{\epsilon^2}$ instances, the expected deviation from the empirical running time is at most $\epsilon$. In summary, we can use PAC bounds to translate the empirical performance observed on sampled instances into theoretical bounds on the expected performance on future instances.

---

> > ### Comment · Reviewer_YN2Q · 2022-08-07
> > **Thanks for your comments**
> >
> > Thanks for your answer, and
> > I look forward to reading "An example of improving the upper bound with special heuristics," which may appear in the final version. Initially, I thought this result was pessimistic, but there seems to be some room for practical impact.

---

### Meta-Review · Area_Chair_xSvx · 2022-08-28

**Recommendation:** Accept
**Confidence:** Less certain

**Metareview:**

Strong paper studying the sample complexity of learning heuristic functions for GBFS and A*. The reviewers were especially impressed with the theoretical results and find the paper a worthwhile contribution to this conference.

**Award:**

No

---

### Decision · Program_Chairs · 2022-09-14

Accept